# Assessing the Outreach of Targeted Development Programmes—A Case Study from a South Indian Village

Anugu Amarender Reddy [1,*] , Anindita Sarkar [2] and Yumiko Onishi [3]

1 Indian Council of Agricultural Research-Central Research Institute for Dryland Agriculture, Hyderabad 500059, India
2 Department of Geography, Miranda House, University of Delhi, Delhi 110007, India; anindita.sarkar@mirandahouse.ac.in
3 IC Net Limited, Delhi 110091, India; onishi@icnet.co.jp
* Correspondence: amarender.reddy@icar.gov.in; Tel.: +91-70423-61439

**Abstract:** This paper explores beneficiary targeting of government programmes in a village in India. The analysis is based on all 228 households of the village and focus group discussions. The results show that there is a large exclusion error in targeted programmes, which have mostly excluded the poor and the needy. Most schemes have a prerequisite of asset ownership, such as agricultural land, which benefits resource-rich farmers with large landholdings. The relationship between benefits received and income of households is best represented by an inverted 'u'-shape curve, indicating the middle-income category benefits more than the poorest. The scope and scale of welfare programmes, especially Direct Benefit Transfers, increased during the COVID-19 pandemic. For inclusion of the poorest of the poor, welfare and development schemes need to be decoupled from landownership in rural areas.

**Keywords:** development programmes; welfare schemes; beneficiary; targeted schemes; socially disadvantaged groups

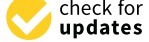



## 1. Introduction

Ending all forms of poverty across the globe is the first goal of the 2030 Sustainable Development Goals (SDGs). Historically, the Universal Declaration of Human Rights of 1948 recognizes the right to social protection in order to make economic growth inclusive. Though the concept of a social safety net dates back to political economists such as Adam Smith, Condorect, and Turgot [1], with the impending deadline of the SDG target, the time has come to look at the performance of poverty alleviation policies and assess if they truly target and benefit the poor. In response to this, governments across the world are increasing their welfare and development budgets. Similarly, the central and local governments in India have also initiated a plethora of programmes and schemes, mainly to address the needs of the poorer households in the society. Some of these schemes are universal, such as a subsidy on cooking gas, a midday meal scheme for school children, pre- and postnatal care and assistance for women etc., while some specifically target households below the poverty line, such as distribution of food grains at subsidised rates.

Poverty is an extremely complex phenomenon, and it manifests itself in a range of overlapping and interwoven economic, political, and social deprivations [2,3]. The programmes and policies aimed at overcoming deprivation and alleviate poverty need to assess "who" is poor and "why" they are poor. Such programmes also require thoughtful investments both in terms of "how much" to invest and in "how" to invest. Hence, targeted development programmes need to carefully select beneficiary households or individuals to both deliver the maximum impact and to optimally use the funds allocated for them.

Targeting is especially important in countries such as India that are characterised by tight fiscal situations with a large poor population in absolute terms. Given the higher

levels of poverty, malnutrition, and underemployment, it has become mandatory for the government to provide income support through subsidies, productive employment opportunities, and social security schemes in rural India [4,5]. Such interventions also include development schemes such as health insurance, subsidised treatments, free education for children, pensions for the elderly and widows, surplus land distribution for the landless, and subsidies to purchase fertilizers, seed, and farm machinery [6]. Many of these programmes are supported by multilateral and bilateral donors who strongly hope to contribute to ending poverty in India. Though the central government allocates funds for poverty alleviation, state governments are the key to India's progress on the SDG Agenda, as they are best placed to 'put people first' [7]. To ensure that 'no one is left behind', India needs policies that reach its targeted beneficiaries for optimal resource utilization and fulfilment of its goal of poverty eradication.

This paper makes an important contribution to address the host of practical, ethical, and political concerns with respect to beneficiary identification for targeted development schemes in India. The paper seeks to investigate the effectiveness of targeting mechanisms of various welfare and development schemes in operation in Kunkudupamula Village of Telangana State in India. It seeks to understand whether the investments in development and social welfare schemes actually benefit the socially and economically weaker sections the schemes target more than the well-off sections.

## 2. Literature Survey

There seems to be weak mapping the nature of poverty reduction programmes to methods of targeting across regions of the world. Studies have noted dominance of cash transfer programs in Eastern Europe, Central Asia, and Latin America; universal food subsidies in the Middle East and North Africa; and a mixture of cash and food transfers in South Asia [8–13].

A number of authors have defined targeting efficiency indicators with a view to identify best targeting practices [14], each having its own advantages, drawbacks, and different degrees of relevance and appropriateness depending on the particular socio–economic context or country under which it is implemented [15]. Means testing methods that identify the poor based on direct measurement of income/consumption/standard of living of households or individuals is used for direct cash transfer policies and is particularly popular in Africa and Latin America [16].

Geographic targeting is widely followed in developing countries within certain geographic areas [17]. It is very popular in India, where there is large concentration of poor in rural areas. Demographic targeting aiming to distribute meals and nutritional supplements to vulnerable population such as preschool children, pregnant and lactating mothers, and women of childbearing age from low-income households are popular in South Asia [16,18].

Given that India is struggling with high fiscal deficits and budget constraints, self-targeted schemes are preferred over universal schemes. In these two types of targeting, (i) individuals recognise the need to be beneficiary or (ii) the local administration/authority identifies the beneficiary. For example, under the Mahatma Gandhi National Rural Employment Act (MGNREGA—the largest employment guarantee programme in the world, which guarantee 100 days of casual employment to individuals who opt to work in their own villages), individuals recognise the need to be beneficiary. Generally, it is assumed that only poorer households opt to work under this scheme as it provides only casual, unskilled work with a low wage. However, because of its universal accessibility, low work intensity, off-hours (from 6 a.m. to 10 a.m.) and transient nature, many non-poor individuals take part [19]. Hence, self-selection gives people the chance to decide for themselves whether they work under an employment guarantee scheme or not, irrespective of their poverty status [20]. On the other hand, under the Public Distribution Scheme (PDS), the government/administration identifies beneficiaries based on their income status. Under this scheme, only households below the poverty line are eligible to receive subsidised food grains.

Despite being one of the fastest growing economies in the world, India is home to the largest number of poor people in the world, making poverty reduction a challenge [21]. It is now established that trickle-down effects do not work in developing countries such as India [22]. Development has remained confined to urban areas [23], and rural areas, where more than three-fourths of the country's poor reside, remain deprived. A deep-rooted caste hierarchy also reinforces social distance between the privileged and the rest [24]. Today, the socially disadvantaged and marginalised population still suffers from hunger, poverty, and deprivation [25].

## 3. Methodology

The study was conducted in Kunkudupamula Village. As India has 600,000 villages, due care was taken while selecting this village so that it would represent an average village in India in terms of population size, landholding, socio–economic composition of population, share of nonfarm activities, and economic activities. The study followed a mixed method approach. All 228 households in the village were selected for an intensive field survey, and a list of farmers was taken from the 2011 village census, with additions (if a household immigrated after 2011) and deletions (if a household emigrated after 2011) were done to cover all the households residing in the village as of the interview date (Table 1). A structured household questionnaire including the general summary of the household, the land and agricultural profile, various household liabilities (including loans), and details of benefits received through various government schemes was developed to understand the linkages between household socio–economic characterization and benefits received by each member of the household. To compliment the analysis, focus group discussions and interviews using semi-structured questionnaires were conducted to understand the actual benefits received by beneficiaries and reasons for exclusion. The semi-structured questionnaires covered various aspects such as the evolution of cropping systems, predominate crops, cost structures, levels of tenancy, tenancy rates, credit availability to households, interest rates and repayment culture, labour markets, frequency and causes of out- and in-migration, employment opportunities other than agriculture, the status of youth and women in the village, scheme effectiveness/utility, the level of corruption, the process of filling up application forms, frequency of visits the local office to enroll as a beneficiary, prevalence of benefit usage, and ways in which people had used benefits. These semi-structured questionnaires were completed by interviewing village elders, educated teachers, agricultural officers, rural development officers, etc. Further, separate focus group meetings were organized among farmers, women, youth, and scheduled caste and tribe households, as within the village, women, youth, and scheduled caste (socially disadvantageous group) households are not able to tell their problems openly in larger groups in front of males, parents, and forward caste (socially advantageous) households, respectively. The key opinions of these groups are presented while discussing the results.

**Table 1.** Demographic details.

| | Category | No. of Households | Average Family Size | Average Years of Schooling by Head of Household | Average Age of Head of Household | Land Holding (Acres) |
|---|---|---|---|---|---|---|
| Social Group | SC | 47 | 3.6 | 2.7 | 56 | 1.2 |
| | OBC | 118 | 4.2 | 3.6 | 53 | 2.7 |
| | FC | 63 | 3.8 | 5.1 | 56 | 4.4 |
| Landholding | Land less | 79 | 3.5 | 4.4 | 52 | 0.0 |
| | Marginal | 46 | 3.8 | 1.7 | 59 | 1.2 |
| | Small | 54 | 4.3 | 4.8 | 51 | 3.3 |
| | Medium | 44 | 4.3 | 3.2 | 59 | 7.2 |
| | Large | 5 | 5.8 | 7.0 | 52 | 19.8 |
| Poverty | BPL | 97 | 5.2 | 7.7 | 51 | 1.95 |
| | APL | 131 | 3.0 | 6.4 | 57 | 3.50 |
| | Total | 228 | 4.0 | 3.8 | 55 | 2.8 |

Source: Data collected and compiled from field survey in Kunkudupamula Village. Note: 1. Government of India classifies some of its citizens based on their social and economic condition, such as Scheduled Caste (SC), Other Backward Class (OBC), and forward caste (FC), or others who are not in any category. The SCs are among the most disadvantaged. 2. The study was conducted in year 2016–2017; during this period average exchange rates were INR 20.65 = USD 1.00 purchasing power parity (PPP). USD 1.90 PPP per capita per day poverty line is equivalent to INR 14,319.39 per year. Based on this, all households falling below this threshold were classified as below poverty line (BPL) or poor, and above it were classified as above poverty line (APL).

## 4. The Study Area and Development Schemes

Kunkudupamula is a village located in the Nalgonda District of Telangana State in India (Figure 1). With a population of 903 and a geographical area of 546 ha, this remote village is located 60 km from the district headquarter. Though the village is connected by all-weather roads, transport facilities remain very poor, with the nearest railway 10 km away. The village has electricity and a mobile network. There is no postal or courier service. There is only one primary school. There is no primary health care centre. It has a basic maternity and childcare centre (Anganwadi Centre). There is a fair-price shop where people can buy subsidised food items as allocated in the PDS.

The low level of education (Table 1) among the villagers manifests as a pool of unskilled and semi-skilled workers. This is reiterated in the occupation structure of the respondents, most of whom are cultivators or agriculture labourers (Table 2). Agriculture is the primary occupation in the village. Although the average landholding of the FCs is significantly larger than that of OBC and SC households (Table 1), income does not vary significantly between them (Table 2). This may be due to significant earnings made from secondary occupations, mostly livestock rearing. Since the 2000s, opportunities from other nonfarm sources, such as construction, services (i.e., watchman, drivers), factories, and petty business have also contributed to additional income. Nevertheless, most SC families in the village continue to remain poor. Families belonging to the OBC community are slightly better off, and those of the FC community remain resource-rich with higher levels of household income. The average income level of the village is slightly below the national average and international standards of poverty (USD 1.90 PPP/capita/day) [26].

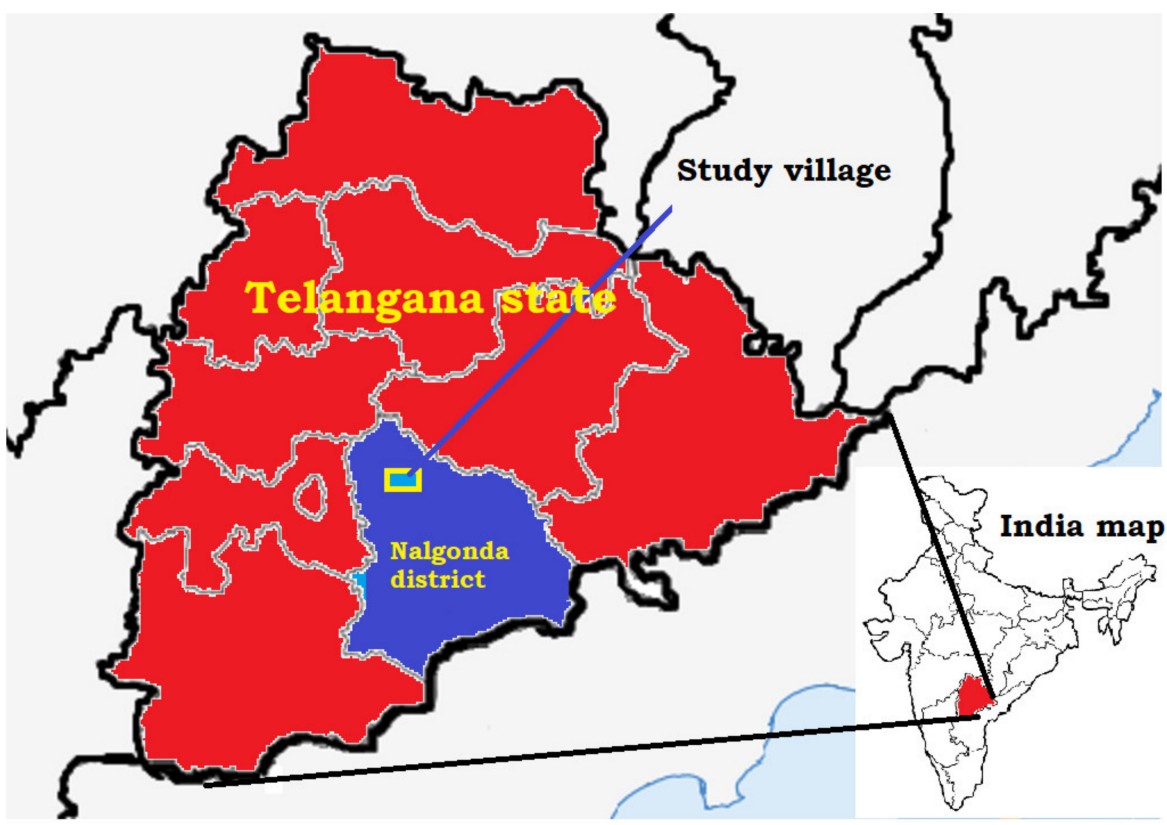

**Figure 1.** Location of study village (Kunkudupamula).

**Table 2.** Distribution of workers by main occupation (averages in percent).

| Economic/Social Categories | Occupation | | | Per Capita Income | |
|---|---|---|---|---|---|
| | **Agriculture** | **Agricultural Labour** | **Non-Agriculture** | **INR per Year** | **USD PPP per Day** |
| **Caste and social category** | | | | | |
| OBC | 64.4 | 25.4 | 10.2 | 18,125 | 2.40 |
| FC | 68.3 | 23.8 | 7.9 | 24,676 | 3.27 |
| SC | 66.0 | 29.8 | 4.3 | 17,509 | 2.32 |
| **Land holding category** | | | | | |
| Landless | 12.7 | 63.3 | 24.1 | 17,456 | 2.32 |
| Large | 100.0 | 0.0 | 0.0 | 27,911 | 3.70 |
| Marginal | 93.5 | 6.5 | 0.0 | 18,436 | 2.45 |
| Medium | 93.2 | 6.8 | 0.0 | 24,850 | 3.30 |
| Small | 94.4 | 5.6 | 0.0 | 19,559 | 2.60 |
| **Income and Poverty** | | | | | |
| BPL | 58.8 | 27.8 | 13.4 | 10,857 | 1.44 |
| APL | 71.0 | 24.4 | 4.6 | 26,436 | 3.51 |
| Total | 65.8 | 25.9 | 8.3 | 19,808 | 2.63 |

Source: Data collected and compiled from field survey in Kunkudupamula Village.

A number of welfare schemes are run by the state; however, to contextualise our findings we will focus on the schemes availed by our respondents in the study village. The welfare schemes availed in the study village can be broadly divided into three types.

First are the schemes that target farmers for increasing agricultural productivity and supporting their incomes. Among these schemes, free agricultural electricity, seed and fertilizer subsidies, soil health cards, and farm loan waiver schemes target all farmers, while the remainder demographically target economically and socially disadvantaged groups among the farmers. The state supplies electricity to all farmers free of cost to run electric pumps to irrigate crops. The state gives INR 4000 (USD 52) per acre twice a year to all the farmers to cover the costs of major inputs such as fertilisers, seeds, and pesticides. The government, in order to free all the farmers trapped in perpetual indebtedness, waived outstanding agricultural loans. However, this scheme covers only institutional loans and does not cover loans from non-institutional sources, such as money lenders. Soil health cards are a central government scheme that assists farmers to scientifically test soil samples from farmlands to help with cropping decisions and input use. Rashtriya Krishi Vikas Yojana provides tarpaulins to farmers on a 50% subsidy to cover their crops and grains.

Second, there are schemes that specifically target women who are vulnerable due to their special needs. To prevent malnutrition, pregnant and lactating women are provided one full meal consisting of rice, lentils, and vegetables for a minimum of 25 days, and a boiled egg and 200 mL of milk for a month at the Anganwadi Centre, along with iron and folic acid tablets, locally known as the Arogya Lakshmi Scheme. This scheme also provides facilities for health check-ups and immunization to pregnant and lactating women. It aims to reduce infant mortality, maternal mortality, incidence of low birthweight babies, and anaemia among women. Another scheme called the Kalyana Lakshmi Scheme gives economic assistance to all newly married brides with a minimum age of 18 years belonging to SC, ST, and minority households. In families with limited resources, child marriage is often seen as a way to provide for their daughter's future. This scheme claims to prevent early marriages and to enhance the literacy rate among girls since the money cannot be claimed for marriages of girls who are younger than 18 years. Abhaya Hastam is a special insurance and pension scheme for women who are above the age of 65 years and were active members of self-help groups (SHGs).

The third type of schemes are general welfare for poor families, such as wellbeing, income, and livelihood support. The MGNREGA programme offers 100 days of manual wage employment in a year in rural areas to every household whose adult members volunteer to do unskilled manual work. Both men and women can work at a daily wage of INR 245 (USD 3.2) per day. The PDS provides basic food items (rice, wheat, sugar, kerosene, etc.) to rural ration cardholders at fixed prices through a network of Fair Price Shops. Beneficiary families with income under INR 150,000 (USD 1972) are given 6 kg of rice per person without any ceiling on number of members in the family at INR 1 per kg. Aarogyasri is a medical insurance scheme that provides BPL families with quality medical care for treatment of identified diseases involving hospitalization, surgeries, and therapies through an identified network of healthcare providers. The entire premium is shouldered by the government. The gas subsidy scheme provides 12 subsidised LPG cylinders per year to all households. The subsidy amount is directly transferred to the beneficiary's bank account. To achieve universal sanitation coverage and "an open defecation free India", the state has launched the Swachh Bharat Telangana Mission that offers poor rural households financial help to construct latrines.

In addition, there are some programmes targeting population with specific needs. To support the traditional shepherd communities known as Kurumas and Yadavas in Telangana, distribution of sheep on subsidy was launched, where every person older than 18 years old from shepherd community is given a 75% subsidy on a unit of sheep, which consists of 20 sheep and a ram. The Aasara Pension Scheme provides financial assistance for the daily minimum needs of the economically weaker sections in the state, aged between 58 to 65 years, widowed, destitute, AIDS patients, weavers, and needy tribal people. A fee reimbursement scheme gives 100% financial assistance to students belonging to poor SC, ST, and "BC" (sic) families enrolling in higher education.

## 5. Results and Discussions

### 5.1. Number of Beneficiaries and Amounts Disbursed

Among the programmes currently being implemented in the study village, the highest percentage of beneficiaries is noted for the gas subsidy programme (Figure 2). This programme follows a universal targeting method, and all households that use LPG cylinders for cooking benefit. Though many respondents claimed that the subsidy is meagre, the scheme indirectly promotes the health and wellbeing of women, as they are almost universally responsible for cooking for their families. LPG cylinders support gas ovens, which are smokeless and save time and effort in cooking. The second largest in terms of beneficiary reach is the PDS. The villagers are very satisfied with the system as they can conveniently avail all the allocated rationed items within the first half of every month.

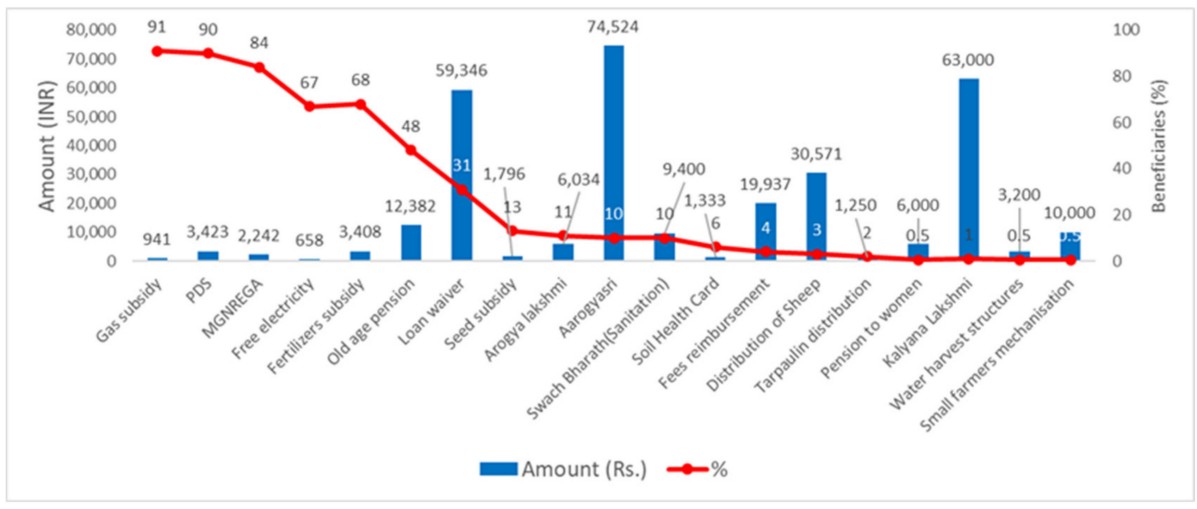

**Figure 2.** Beneficiary households (% of total) under different schemes and average amount.

The other programmes with large beneficiary groups are the self-targeted MGNREGA and demographically targeted schemes that distribute subsidised agricultural inputs to farmers. Although there were mixed responses about the procedure and timeliness of wage reimbursements, most people in the village who worked in MGNREGA did not have their own farm or business. Thus, it has definitely helped the unemployed and the landless, and has kept a competitive wage rate for manual labour in the village. Similar studies have also noted a higher percentage of the poor participating in and benefitting from MGNREGA [27–29], as it enhances their purchasing power [30]. At the same time, many non-poor individuals also take part in such programmes [19], but in our study village this was not the case. Instead, some farmers we interviewed raised concerns over shortages of agricultural labourer and rising wage rates. This situation is a result of casual labourers opting to work in MGNREGA and demanding higher wages to work on private farms for agricultural operations. These farmers suggest that wage payments for agricultural operations on private farms should also be eligible to get funds under MGNREGA so as to avoid the repercussions from rising wage rates triggered by MGNREGA.

In terms of monetary benefits, the highest disbursement has been for agricultural loan waivers for farmers, benefiting around 30% of surveyed households (Figure 2). Small outstanding loans up to INR 25,000 are waived in the first phase, followed by larger amounts, with a maximum limit of INR 100,000. Some farmers with large loans expressed dissatisfaction with the scheme, and some lamented that it required a lot of paperwork and a long waiting time for disbursement. Furthermore, some farmers complained that the government delayed payments to banks, and as a result, they are unable to take out fresh loans. On the other hand, bankers complained that it affects the repayment culture among farmers, reducing disbursal of fresh loans. The other two programmes giving large pay-outs were Aarogyasri and Kalyana Lakshmi. Despite the fact that Aarogyasri, a health

insurance scheme, has a maximum limit of INR 500,000, the fund is only available for the cost of surgeries and inpatient treatment; hence, beneficiaries still have to bear the cost of medicines and outpatient treatments. In the case of Kalyana Lakshmi, every person who applied for the scheme received the benefit amount, but there was general concern about delays in getting the money. However, the unique thing about this scheme is that it is only available for brides of 18 years and above, and the money was directly transferred to the bank account of the bride's mother, focusing on gender empowerment.

## 5.2. Benefits by Social Groups

If we look at the social category of the beneficiaries of the programmes and the money disbursed (Figure 3A,B), it shows all SC households are covered by the PDS scheme, followed by the gas subsidy (95.6%), MGNREGA (91.3%), and the fertiliser subsidy (67.4%). However, none of the SC category households benefited from water harvesting structures, Kalyana Lakshmi, or the Small Farmer's Scheme. Most probably, SC households could not avail the Kalyana Lakshmi scheme as either they did not have bank accounts or their aughters were married before the age of 18. Some earlier studies have noted that in programmes that involve direct cash transfers to beneficiaries, they do not reach the transient poor in most cases [31], and have the disadvantage of costly documentation [32]. Nevertheless, with the nearly universal bank accounts of households, beneficiaries of this study get timely benefits with less intermediation cost. Many respondents did not know about the water harvesting structure scheme, and many believed there should be more schemes specially targeting SC households.

Of all the programmes, loan waivers account for the largest amount. The FCs benefitted most from this scheme. On average, the amount crop loans waived was INR 61,000, INR 49,385, and INR 32,231 (USD 803, 650, and 425) for families belonging to FCs, "BCs" (sic), and SCs, respectively. Arogyasri also accounted for huge sums of money across all three caste groups, but only for a few households. It involved INR 50,000, INR 48,900, and INR 43,333 (USD 659, 644, and 571) for families belonging to FC, "BC" (sic), and SC, respectively. The subsidy to purchase sheep and the Kalyana Lakshmi schemes benefited the "BC" (sic) families the most, but only a few households benefited from both schemes.

## 5.3. Benefits by Landholding Class

The Telangana government supplies 24 h/day of free electricity to run irrigation pump in the state. It is available to all farmers who have tube-wells fitted with electric motors and electricity connected to their farms (Table 3). Most of the poor who were either landless or marginal or small farmers did not benefit from this input subsidy as they did not own any wells. Earlier studies have also noted that the free electricity subsidy only favours the resource-rich, tube-well owners, furthering inequality among farmers in the villages [33].

More than half of the landless and marginal farmers benefited from Aasara pension's scheme. However, the respondents who benefitted from this scheme complained about irregularity and delays in starting the pension. Some even confided that officers concerned with the pension had to be offered kickbacks to complete the process. Only 2.3% of medium farmers benefited from the water harvesting structure scheme, as there was a lack of water structure construction in the village.

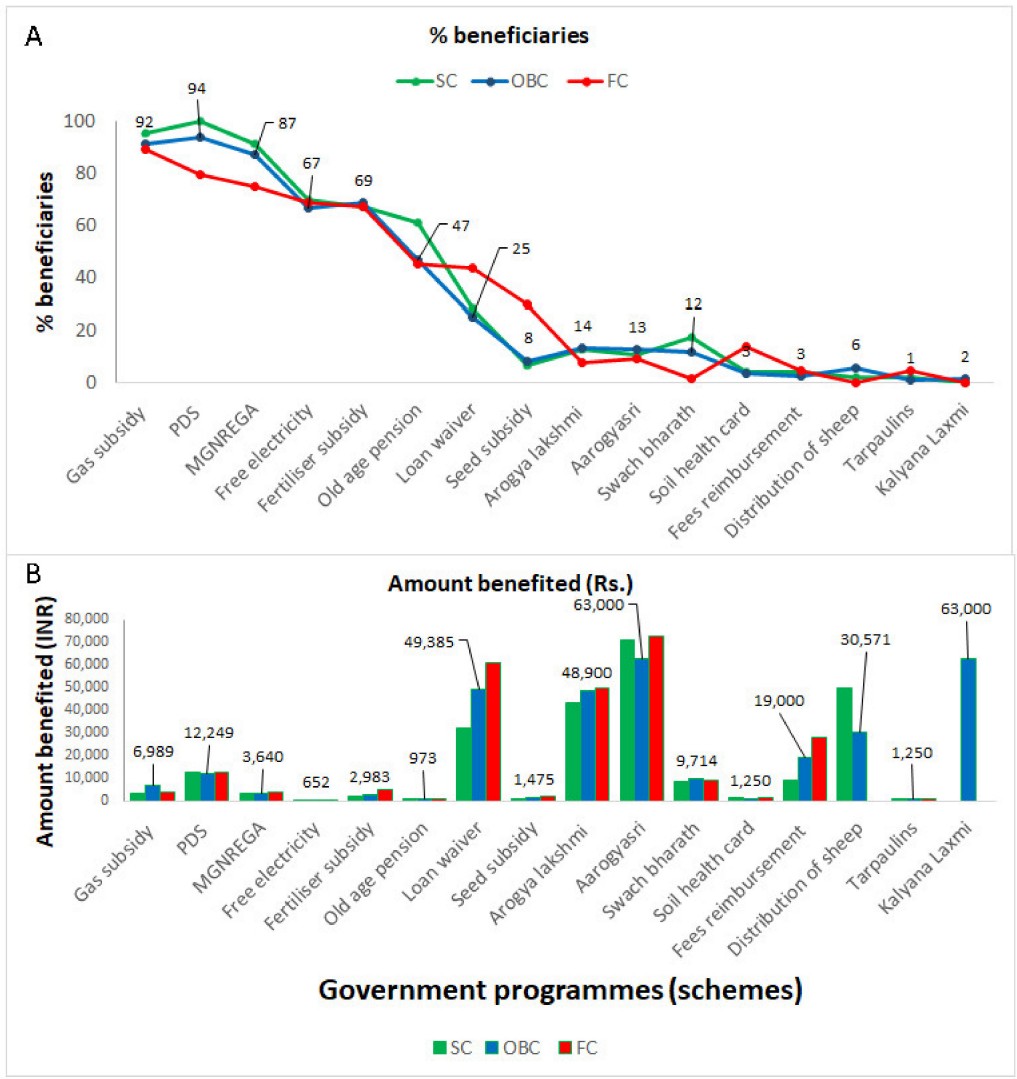

**Figure 3.** (**A**) Beneficiary households (% of total) under different schemes by social group (**top**); (**B**) average amount benefited in Rupees (**bottom**). Note: women's pensions, small farmer's mechanisation, and water harvest structures are not included as the beneficiaries are less than 1%.

**Table 3.** Average amount benefitted by households under different schemes (INR).

| Schemes | Landless | Marginal | Small | Medium | Large |
|---|---|---|---|---|---|
| **Gas subsidy** | 936 (83.5) | 913 (93.5) | 958 (98.1) | 1023 (95.5) | 1100 (100) |
| **PDS** | 3285 (96.2) | 3634 (100) | 3742 (98.1) | 3830 (68.2) | 5376 (60) |
| **MGNREGA** | 2509 (86.1) | 2678 (80.4) | 2325 (98.1) | 2187 (72.7) | 2167 (60) |
| **Free electricity** | 600 (12.7) | 573 (97.8) | 604 (94.4) | 757 (100) | 1080 (100) |
| **Fertilizer subsidy** | 4618 (7.6) | 1448 (100) | 3313 (100) | 4471 (100) | 8163 (100) |
| **Old age pensions** | 12,314 (53.2) | 13,000 (65.2) | 11,783 (42.6) | 12,750 (36.4) | 0 (0) |
| **Loan waiver** | 0 (0) | 32,909 (23.9) | 57,633 (55.6) | 64,692 (59.1) | 100,000 (80) |
| **Seed subsidy** | 0 (0) | 1250 (4.3) | 1521 (25.9) | 1696 (31.8) | 3500 (40) |
| *Aarogya Lakshmi* | 8182 (13.9) | 4365 (10.9) | 40,00 (11.1) | 4000 (6.8) | 3413 (40) |
| *Aarogyasri* | 83,333 (11.4) | 62,500 (8.7) | 65,000 (20.4) | 27,000 (4.5) | 0 (0) |

**Table 3.** *Cont.*

| Schemes | Landless | Marginal | Small | Medium | Large |
|---|---|---|---|---|---|
| *Swach Bharath* (sanitation) | 3000 (1.3) | 9125 (17.4) | 10,500 (18.5) | 8500 (9.1) | 0 (0) |
| **Soil Health Card** | 0 (0) | 0 (4.3) | 1357 (13) | 1333 (13.6) | 0 (0) |
| **Fees Reimbursement** | 12,000 (5.1) | 9250 (4.3) | 46,500 (3.7) | 0 (0) | 0 (0) |
| **Distribution of sheep** | 21,000 (1.3) | 40,667 (6.5) | 30,333 (5.6) | 30,000 (2.3) | 0 (0) |
| **Tarpaulins** | 0 (0) | 1250 (2.2) | 1250 (5.6) | 0 (0) | 1250 (20) |
| **Women's pension and insurance** | 0 (0) | 6000 (2.2) | 0 (0) | 6000 (2.3) | 0 (0) |
| *Kalyana Lakshmi* | 0 (0) | 0 (0) | 75,000 (1.9) | 51,000 (2.3) | 0 (0) |
| **Water harvesting structures** | 0 (0) | 0 (0) | 0 (0) | 3200 (2.3) | 0 (0) |
| **Small farmers mechanization** | 0 (0) | 0 (0) | 10,000 (1.9) | 0 (0) | 0 (0) |

Note: Figures in parenthesis indicate percentage share of households against the total in each landholding category.

*5.4. Relationship between Subsidies and Land Ownership*

Benefits from subsidies and welfare programmes have an inverted 'u'-shape relationship with land ownership (Figure 4). The best fit model is a cubic relationship between land ownership and total welfare benefits, indicating that the landless benefited least. However, benefits were higher for small farmers, but for medium farmers benefits were reduced, and they spiked for large farmers. MGNREGA is the only programme that is universally accessed by the landless. Most of the poverty reduction schemes in rural India are intrinsically directed to support agriculture linked to land and associated assets such as free electricity, crop loan waivers, subsidies for seed and fertilizer, and even direct cash transfer to farmers. Although welfare and development schemes linked to land ownership increase opportunities for agricultural labourers by increasing employment opportunities and raising wage rates, they directly exclude all the landless households.

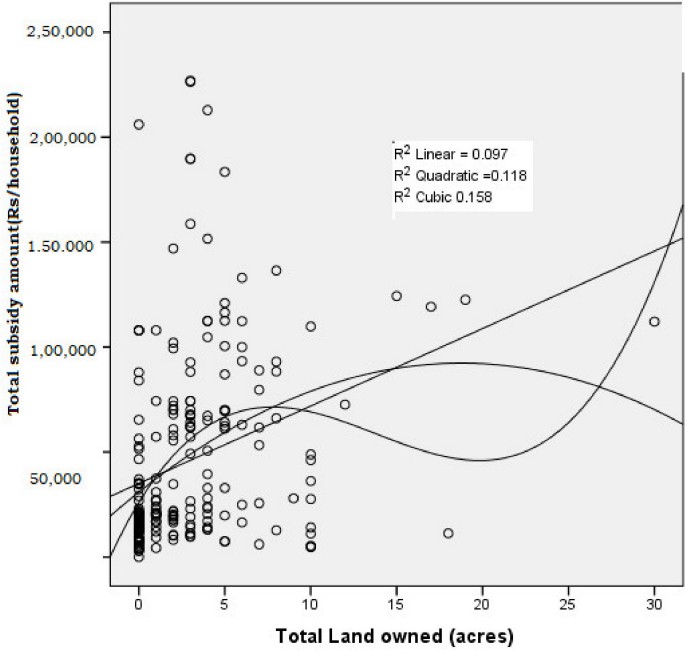

**Figure 4.** Relationship between land ownership and total subsidy.

### 5.5. Relationship between Household Income and Total Subsidy

Since landownership has a strong positive correlation with household income, we also observe an inverted 'u'-shape relationship between the total benefits received by households from welfare programmes and total household income (Figure 5). Similar observations were made in other studies [34,35]. This indicates a large exclusion error, particularly among the bottom 20% of households. In other words, schemes that are launched for rural households to reduce poverty are also being used by the non-poor households in the village, and often the actual poor are not benefiting.

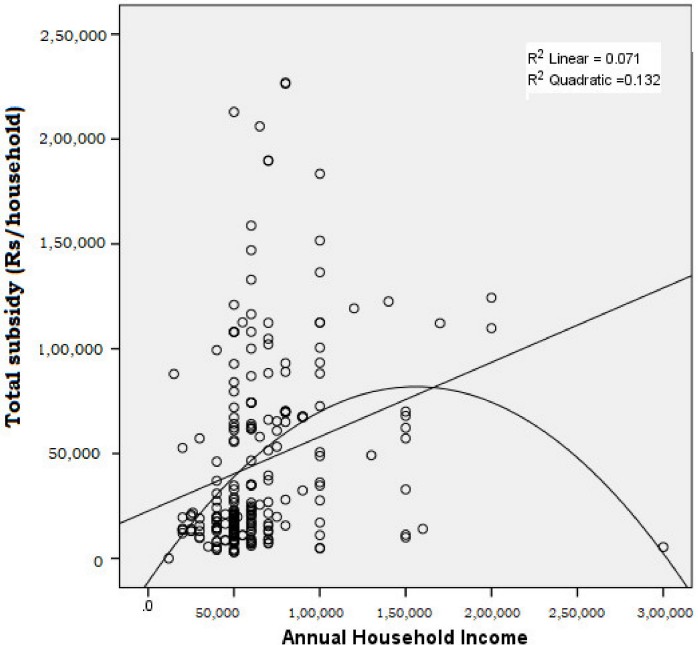

**Figure 5.** Relationship between household income and total subsidy.

Such exclusion errors in targeting have long-term implications. They tend to increase vertical inequality, i.e., inequality between the poor and the non-poor. Simler and Nhate [36], in their study in Mozambique, also found that geographical targeting may not benefit the poor in places that have fairly heterogeneous poverty levels. Thus, in unequal societies, the programmes that aim to benefit the poor need to be reoriented and redesigned in such a way that exclusion does not happen in its targeting.

### 5.6. Relationship between Benefits and Social Class of the Beneficiary

In an empirical analysis [11], Zacharias and Vakulabharanam established (as expected) that the disadvantaged groups in India known as the SC and ST have substantially lower wealth and assets than the FCs. In our study village, too, the families with large land holdings with wealth were high caste. In line with our discussion in the previous section, it is clear that the benefits from the programmes have disproportionately gone to the rich, land-owning FC households (Figure 6). FC households were getting the highest public subsidy, with INR 55,660 (USD 733), followed by the "BCs" (sic), with INR 43,160 (USD 568), and INR 37,834 (USD 500) for SCs. The average benefits were lowest among labourers, at INR 22,313 (USD 294), and also among the landless, at INR 35,826 (USD 472).

With empirical evidence of large exclusion of the poor and the socially disadvantaged groups from the welfare schemes, our study suggests that social welfare schemes benefit the socially and economically better-off households, further pushing the poor and/or SC families towards poverty. This is inevitable as poverty in India is complex, with historical roots and socio–economic inequality due to skewed ownership of landholdings in favour of high castes. As discussed in the earlier section, the farmers with large landholdings

benefit more from the government programmes due to the large sums of institutional loan waivers, free electricity with tube-well ownership, and fertilizer subsidies. These benefits also increase with increased farm size.

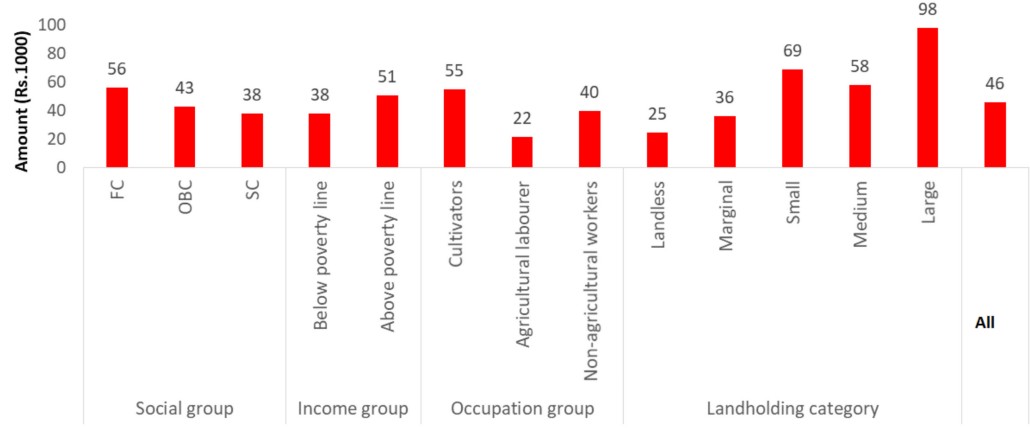

**Figure 6.** Benefits accruing to different beneficiary groups under different schemes.

### 5.7. Utilization of Benefits

Among the utilization of benefits, free electricity, the fertilizer subsidy, and soil health card schemes were accessed by all households (Figure 7). The schemes that were linked to agricultural development were inadvertently used by landowning households and did not cater to the general welfare of the poor who were landless or to agricultural labourers.

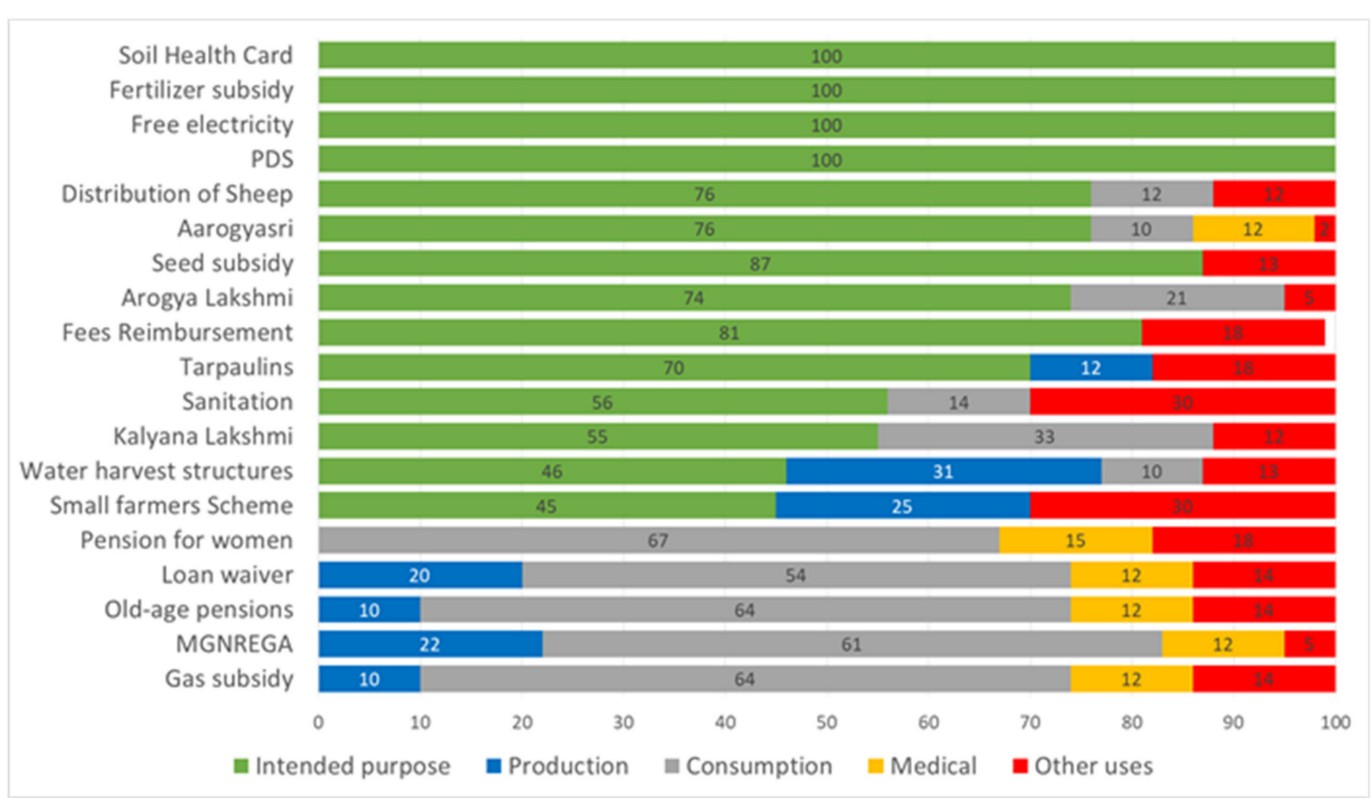

**Figure 7.** Utilization of programmes by purpose (% of beneficiaries).

### 5.8. Are the Programmes Targeting the Poorest of the Poor?

Our household survey data were divided into two income categories based on the annual income of the respondents: the bottom 20% of households are called the poorest-

of-the-poor, and the other 80% are referred to as not-so-poor. Our analysis notes that the benefits accruing to the not-so-poor households are systematically higher than benefits received by the poorest-of-the-poor households (Figure 8A,B). This indicates targeting errors and loopholes in implementation.

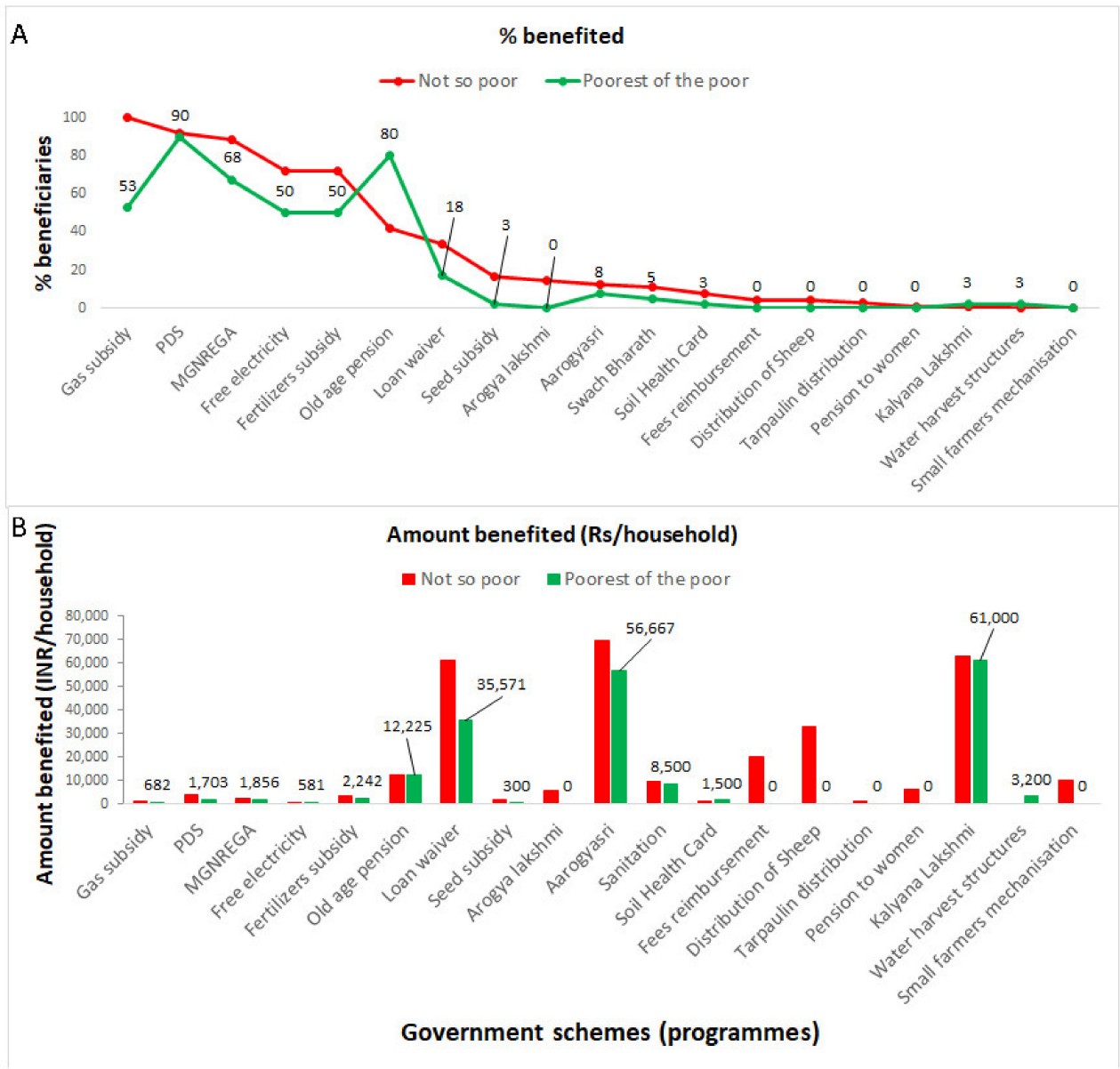

**Figure 8.** (**A**) Beneficiary households (% of total) covered under different schemes and (**B**) average amount.

We observed large exclusion errors or F-mistakes for "Failure to cover" in the welfare schemes in our study village. Such errors occur when targeted persons do not benefit from the very programmes meant for them [37], in this case, the "poorest of the poor". Most of the schemes do not reach the poorest of the poor because they come with a precondition of asset ownership, which the poorest do not have due to their poverty. Thus, the programmes often fail to target, leading to exclusion errors. Table 4 summarises the targeting errors and their reasons for each programme.

**Table 4.** Targeting errors and gender priority of development and welfare programmes in the study village and their reasons.

| Programmes | Description | Type of Targeting | Targeting Errors | Reasons for Poor Targeting of Beneficiaries | Gender Rating (Min = 0; Max = −5) |
|---|---|---|---|---|---|
| Gas subsidy (cooking) | To provide 12 subsidized LPG cylinders per year to households | Universal targeting | Large inclusion error | This universal scheme benefits all, but the poorest of the poor are deprived due to their inability to purchase gas stoves. | 5 |
| PDS (Public Distribution System) | The Indian food security system to distribute food grains (mainly rice and wheat) to India's poor at subsidised rates | Targeted to below-poverty-line households | Large exclusion and inclusion error | No proper income estimations | 3 |
| MGNREGA | To guarantee 100 days of employment for every household per year through arranging public works in the vicinity of the village | Self-targeted workfare | Small exclusion error | Due to sickness and physical disability, some poorest of the poor are not able to work. | 5 |
| Free electricity | Free electricity for irrigation to run motors of pump sets for agriculture | Targeted to farmers with bore well and electricity connection | Large exclusion error | Only accessible to rich, landowning households having tube-wells connected with electric motors. | 0 |
| Fertiliser subsidy | To make fertiliser application affordable to farmers to increase yields | Targeted to farmers who use fertilisers | Exclusion error | Farmers with large landholdings and access to irrigation buy more of these fertilizers, while small landholders get little benefit and landless labourers do not get any subsidy | 0 |
| Old age pension | Aasara pensions (old age pension): A type of pension provided to elderly, disabled, widows, elderly weavers, and other informal sector workers | Demographic targeting | Exclusion error | Some elderly do not get pensions due to lack of supporting documents such as proof of age or residence certificate | 3 |
| Loan waiver scheme | To waive crop loans taken from formal financial institutions (scheduled commercial banks) up to 1 lakh | Targets farmers taking loans from formal sources such as banks | Exclusion error | Poor farmers without collateral who take loans from informal sources and tenant farmers get excluded. | 0 |
| Subsidy on seed | To supply quality, improved, and certified seed to farmers at 50% subsidy | Targeted to farmers who buy seed from government sources | Large exclusion error | Only paddy farmers who purchase seed from the government are getting seed subsidy. All private seed is without seed subsidy. | 1 |

**Table 4.** *Cont.*

| Programmes | Description | Type of Targeting | Targeting Errors | Reasons for Poor Targeting of Beneficiaries | Gender Rating (Min = 0; Max = −5) |
|---|---|---|---|---|---|
| Aarogra Lakshmi | Aarogya Lakshmi is a nutritional program to support pregnant and lactating women. The scheme is available for women below and above poverty line | Pregnant and lactating women | Some exclusion error | Some pregnant women are not enrolled in the scheme | 5 |
| Aarogyasri | To cover the health expenses of the poor in case of hospitalization | Targeted to in-house hospitalisation | Large exclusion error | Restricted only to hospitalised people. | 3 |
| Swach Bharat | To maintain cleanliness and eradicate open defecation, government subsidies for construction of toilets in their houses for poor households | Universal targeting | Some exclusion error | Only households with own house are eligible for subsidy | 5 |
| Soil Health Card (SHC) | Although farmers are aware of the SHC scheme, until now farmers of the village did not have the SHCs. As the villages are remote and affected by Maoist movement, agricultural officers are reluctant to go to the villages to collect soil samples. | Universal targeting of all farmers | Large exclusion error | Many farmers have not received the SHCs | 2 |
| Fee reimbursement | To provide scholarships to the students of economically weaker sections pursuing higher education | Targeted to cover higher education fees of below-poverty-line households | Exclusion error | Due to poor educational facility in the village, poor children drop out after primary schooling. Only high-caste rich people can afford to enrol their children in higher education. | 2 |
| Distribution of sheep on subsidy | To provide traditional shepherd families 20 + 1 sheep on 75% subsidy for their development to increase livelihoods from livestock rearing | Targeted to shepherds | Exclusion and inclusion error | Does not cover all households due to limited budget and also no distribution of sheep to non-shepherds. | 3 |
| Tarpaulin (under Rashtriya Krishi Vikas Yojana: RKVY) | Tarpaulins are distributed to farmers for covering grain from rains, use as drying platforms, used for cleaning and grading of grains after harvest. | All small holding farmers | Balanced | Only small fee is required for each tarpaulin, hence most farmers benefited. | 2 |

**Table 4.** *Cont.*

| Programmes | Description | Type of Targeting | Targeting Errors | Reasons for Poor Targeting of Beneficiaries | Gender Rating (Min = 0; Max = −5) |
|---|---|---|---|---|---|
| Abhayahastam | SHG women contribute INR 365 per annum and government co-contribution of INR 365 per annum into pension account. Interest is earned on the principal to age 60 years, then is paid out as monthly pension | All women Self-Help Group (SHG) members are targeted group | Non-SHG members are excluded. | Many non-SHG women are not covered | 5 |
| Kalyana Lakshmi | Help girl's families with financial assistance of INR 75,000 to cover marriage expenses. | All households with girls age above 18 years are covered | No inclusion or exclusion error | | 5 |
| Water harvesting structures | Subsidies for construction of rainwater harvesting structures on farmland | Targeted to all farmers | Exclusion error | Only a few farmers with large landholdings benefit | 3 |
| Small farmers scheme | Assistance to small landholding farmers to purchase small inputs such as plough and sickle | Targeted to small farmers | Exclusion error | Due to limited budget allocation, very few benefit | 3 |

Note: Gender rating 1–5 scale is developed to understand gender focus of different schemes: 1 = no focus on women, 2 = less focus, 3 = medium focus, 4 = more focus; 5 = 100% focus.

It has been observed that large exclusion errors reduce programme costs but diminish impact in poverty alleviation [15]. By transferring resources to non-poor individuals, the programmes increase vertical inequality, i.e., between the poor and the non-poor [14], impeding horizontal efficiency and creating resentment and social instability [38]. Many studies have also shown that the risk of capture by local elites grows with inequality in land and asset ownership [33,39] and with resource depletion [33], particularly in programmes that intend to target poor farmers for agricultural input and credit distribution. Unfortunately, local governments in areas with the highest inequality tend to choose less-efficient targeting systems [33,40], as in societies with greater inequality, the poor have less of a voice in decision-making processes [14]. Such beneficiary selection mechanisms also tend to perpetuate local power structures and exclude certain individuals from the welfare programmes for social and ethnic reasons [41].

Exclusion error in targeting of beneficiaries is a result of other factors as well, such as lack of transparency in identification of beneficiaries, implementation of schemes and associated administrative procedures, as well as lack of manpower and/or technical capacity of government personnel at the field level for monitoring and social mobilization.

*5.9. Income Inequality and Reach of Subsidies*

In the village, inequality plays an important role in every life and in socio–economic relationships between households. In the village, inequality in landownership is high, further inequality in irrigated landownership is much higher, but income inequality is less. This is mainly due to non-land-based incomes, such as casual labourer, nonfarm and off-farm employment, business, and self-employment opportunities, in villages have increased over the past two decades. The reach of subsidies such as gas and food subsidies is more equal among village society, as they are not based on assets (particularly land); however,

some land-based subsidies, such as credit subsidies and agricultural input subsidies, such as fertilizer subsidy, are highly skewed (Figures 8 and 9).

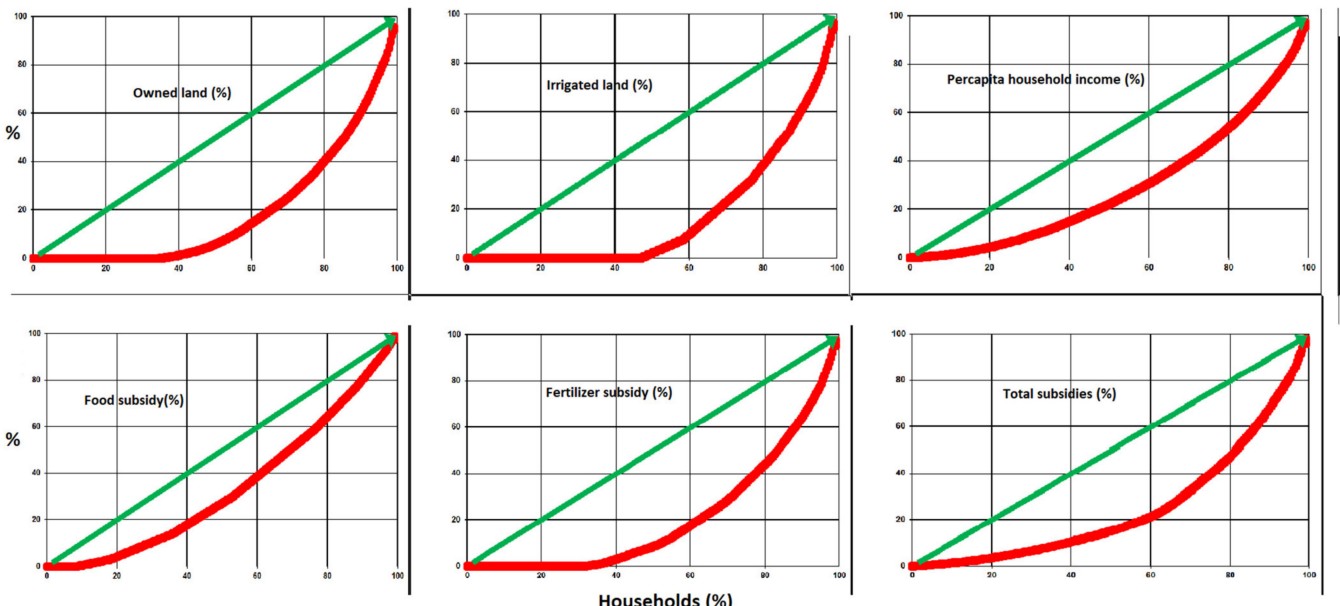

**Figure 9.** Lorenz curve for landownership, irrigated area, per capita income, food, fertilizer, and total subsidy.

*5.10. COVID-19 Shock and Agility of Government Schemes during Pandemic*

India reported its first COVID-19 case in March 2020, and there were three waves between 2020 and 2022 (Figure 10). All countries pumped a lot of money towards social safety nets to protect households from job losses and for income opportunities, including India. COVID-19 is the test case for understanding the agility of government schemes towards external shocks through various social safety nets. With the increased focus on direct money transfer since 2013, the systems from top to bottom are all strengthened to transact huge sums of money. As part of overall measures to increase transparency, the government also promoted direct benefit transfer (DBT) in all government schemes (Figure 11). In 2022, some 420 schemes are working in DBT to reach beneficiaries. The DBTs were facilitated by Pradhan Mantri Jan Dhan Yojana (the Prime Minister's People's Wealth Scheme), which is a financial inclusion program of the government of India open to Indian citizens that aims to expand affordable access to financial services such as bank accounts, remittances, credit, insurance, and pensions. Under this scheme, almost all households have Jan Dhan bank accounts, including the poorest of the poor in India. Further, DBTs are accelerated by the Jan Dhan–Aadhaar–Mobile (JAM) trinity, which refers to the government of India's initiative to link Jan Dhan bank accounts, mobile numbers, and Aadhaar cards (national unique identity card) of Indians to target beneficiaries and plug the leakages of government benefits.

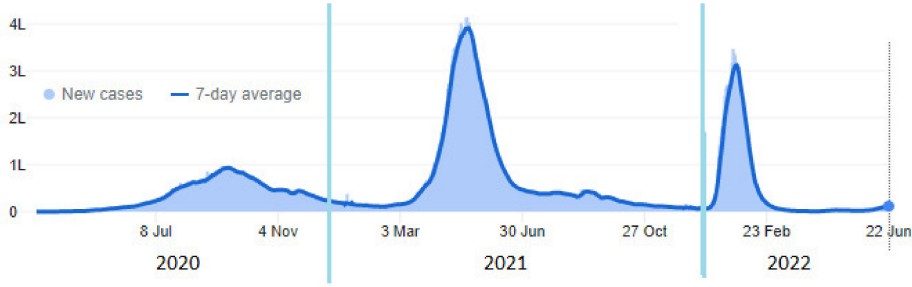

**Figure 10.** Corona cases in India from 2020 to 2022. Source: https://github.com/CSSEGISandData/COVID-19 (accessed on 20 June 2022).

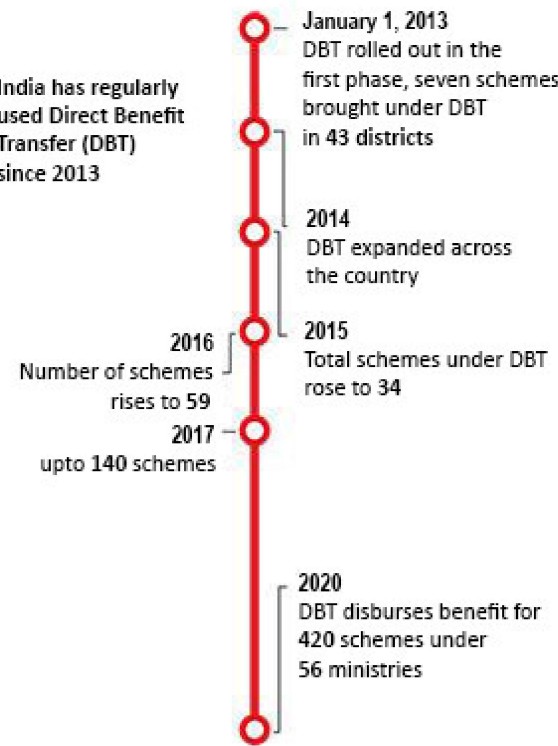

**Figure 11.** Progress of Direct Benefit Transfers (DBTs) in India. Source: https://www.nic.in/blogs/direct-benefit-transfer-a-blessing-during-the-time-of-pandemic/ (accessed on 20 June 2022).

Figure 12 shows the number of beneficiaries under DBT in millions under different schemes for financial years 2016–2017 and 2021–2022. The years 2020–2021 and 2021–2022 fully reflect the additional direct money transfers due to the COVID-19 pandemic outbreak. If we compare the COVID-19 year 2020–2021 with the normal year 2018–2019, there is a huge jump in the number of beneficiaries in the PDS as the government pumped in more free food grains to ensure food security to all households, including migrants. There is also a huge increase in the employment guarantee programme (MGNREGA), as many casual labourer who lost employment in urban areas returned to villages and engaged in MGNREGA public works.

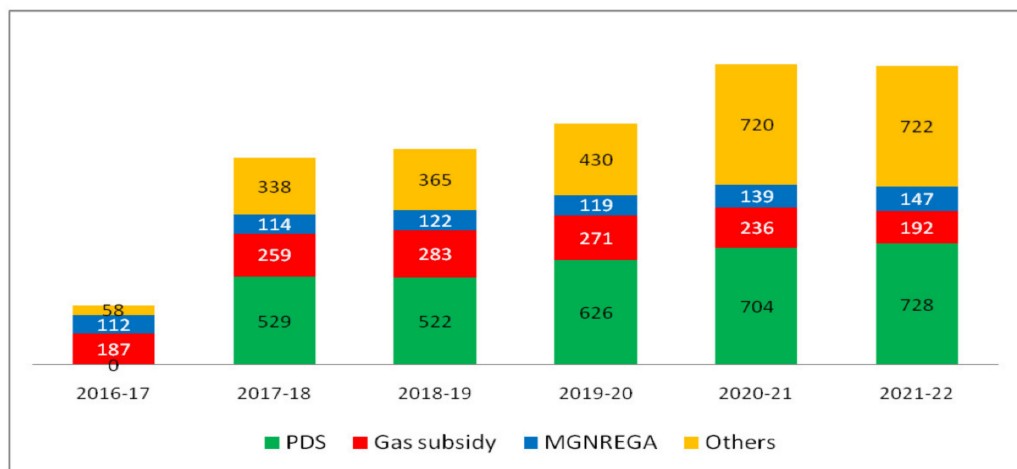

**Figure 12.** DBT beneficiaries under different schemes (million). Source: https://dbtbharat.gov.in/ (accessed on 20 June 2022).

Figure 13 depicts the amount of money transferred through DBT pre- and post-COVID. There is again a huge increase, particularly in PDS, MGNREGA, and even the fertilizer subsidy, as global fertilizer prices increased. In order to compensate for this, the government hugely subsidised fertilizers.

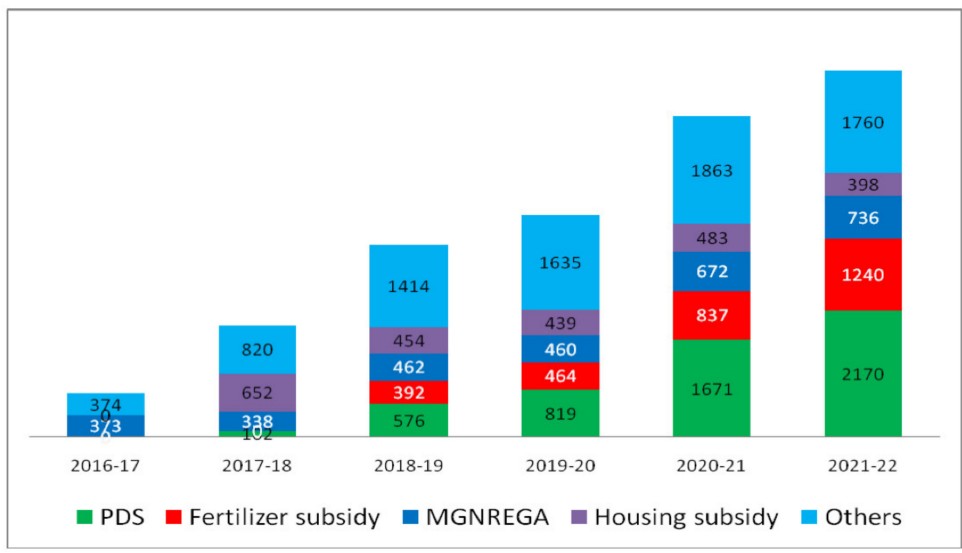

**Figure 13.** Direct benefit transfer (in billions of rupees in cash or kind) through DBT under different schemes in India. Source: https://dbtbharat.gov.in/ (accessed on 22 June 2022).

Overall, there was a huge increase in government support during the COVID-19 pandemic shock, especially in rural areas. The same is reflected in focus group discussions during the COVID-19 period among the households in the study villages.

## 6. Conclusions

Poverty is pervasive in India. However, poverty is mostly concentrated in particular social groups and also in rural areas, where the majority of households depend on agriculture. It has become a social phenomenon where a particular section of society or a social group is unable to fulfil their basic necessities for life [42]. This study intensively examined the reach of various government welfare and development schemes among the poor, socially disadvantageous groups of a population in a village in India by selecting a representative village in the Telangana State. The data were collected by using mixed

methods, with primary data collection from all households along with focus group interactions to understand the dynamics of the functioning and reach of social safety net benefits. All households living in the village were surveyed in order to guarantee inclusion of the poorest of the poor and socially disadvantaged groups so that their opinions will get due representation in the sample. We observed that some universal social safety net programmes such as MGNREGA, PDS, the gas subsidy, and the midday meal scheme benefit the majority of the population, while others benefit the rich more than the poor despite targeting the poor [43]. Our study also had similar findings. The results show an inverted 'u'-shape relationship between total benefits received by households from welfare programmes and household income and land ownership. Hence, there is a need to step up access to welfare schemes for the bottom 20% of households with more-efficient targeting methods.

The eligibility criteria and targeting mechanism employed for identifying beneficiaries under the schemes have significant influence on who will benefit from the scheme. Most of the schemes in our study village targeting rural populations had a prerequisite for asset ownership, such as land. Thus, large landowners who had large institutional loans benefitted from loan waivers and were also irrigating their crops with free electricity and buying subsidised fertilizers. Even the poorest did not benefit from the gas cylinder subsidy as they did not have a gas oven and were still dependent on fireweed for fuel. Without collateral, the poor people had to depend on informal loans and borrowing, thus were not able to benefit from the government loan waiver. Without a secondary school in the village, the poorest were also unable to enrol their children in secondary education and hence were disqualified for the higher education welfare programmes.

It is evident that the success of targeted poverty schemes in India is dependent on proper identification of beneficiaries, transparency, supervision over field staff, and social mobilization. In the past, the central government has allocated fewer resources to villages and regions where poverty, inequality, and the proportion of low-caste individuals are high [39]. Though funds have more than tripled in the last ten years for poorer districts, it is questionable whether there is sufficient capacity at the district and sub-district levels to absorb the funds and produce quality results with existing capacity. Given that some schemes require a lot of procedures and documentation and coordination among various departments, many illiterate households are unable to take benefits even if they qualify for such programmes. Without assessment of existing capacity and in absence of strengthening the very agencies at the field level who have to take on the massive task of delivering a number of schemes, many welfare schemes take off far slower than intended, and often end up being implemented in a haphazard manner—this also includes inappropriate targeting of beneficiaries. This calls for simpler and universal minimum income support for every poor household living in a rural area [44]. Moreover, good governance is the key to successful implementation, and local governments should be incentivised to improve their performance in this regard. The importance of government welfare and social safety net schemes further increase during external shocks such as the COVID-19 pandemic in order to alleviate pain of loss of employment and to increase food security.

However, this paper did not do an intensive field survey post-COVID-19 to understand the actual benefits received by the poorest of the poor during the COVID-19 period. Future studies can focus on dynamics of reach of these social safety and welfare schemes to the bottom 20% of households, especially during external shocks such as pandemic or conflict.

**Author Contributions:** Conceptualisation, analysis, writing, A.A.R.; Review, A.S.; Editing and discussion, Y.O. All authors have read and agreed to the published version of the manuscript.

**Funding:** This research received no external funding.

**Institutional Review Board Statement:** Not applicable.

**Informed Consent Statement:** The authors have taken informed consent from all the households surveyed.

**Data Availability Statement:** Data is available on request.

**Acknowledgments:** The authors thank the farmers for their participation in the survey.

**Conflicts of Interest:** The authors declare no conflict of interest.

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
