# Peer review of "Assessing the Outreach of Targeted Development Programmes—A Case Study from a South Indian Village"

_land, doi:10.3390/land11071030_

Round 1
Reviewer 1 Report
This article is difficult to read and is recommended to be rejected. Specific comments are as follows:
(1) The introduction to the study needs some rewriting. The present introduction does not show the key scientific problems to be solved and the marginal contribution of the research. The author needs to clearly point out these core things in the introduction.
(2) As the key scientific problems to be solved and the marginal contribution are not clear, the author does not clearly define the core concepts in the title of the paper, resulting in chaos in the literature review section, and I do not know what the author wants to express.
(3) As with the previous two opinions, I don't know what the author wants to do, so I can't judge whether the result analysis part is good or bad.
(4) It is suggested that the author find a professional English speaker to edit the whole text systematically.
Author Response
Reviewer comments
Reveiwer-1
This article is difficult to read and is recommended to be rejected. Specific comments are as follows:
- The introduction to the study needs some rewriting. The present introduction does not show the key scientific problems to be solved and the marginal contribution of the research. The author needs to clearly point out these core things in the introduction.
Introduction rewritten
- As the key scientific problems to be solved and the marginal contribution are not clear, the author does not clearly define the core concepts in the title of the paper, resulting in chaos in the literature review section, and I do not know what the author wants to express.
Scientific problem is defined and contribution of the paper is written. Literature updated.
- As with the previous two opinions, I don't know what the author wants to do, so I can't judge whether the result analysis part is good or bad.
Now, changed the entire draft according to the objectives
(4) It is suggested that the author find a professional English speaker to edit the whole text systematically.
English editing is done

Reviewer 2 Report
Dear Authors,
I consider that the article addresses an interesting topic related to the analysis of role of various welfare and development schemes implemented by Indian government to reduce poverty among the population of the Kumkudupamula village.
Introduction:
The aim of the study is mentioned on the literature survey section. You should include it at the end of introduction.
Methodology:
Description of study area: the maps of study village should be inserted in this subsection. I think that the figure 1B is not relevant in the sense that it does not provide aditional information about the location of the village. Moreover, the figure 1b is unclear. If you have the ability to elaborate a map that reflects land use, it would be best to include it-this is a proposal, not a mandatory suggestion.
Can you argue why you choose this case study?
You should present several details regarding the manner of application of the semi-structured questionnaires: Who submitted the questionnaires? The period of application; did you use a model of the questionaire from other studies or do you make a specific structure in relation with the characteristics of the case study?
You refer also to the method focus group but there are no details about the implementation of this method. Which stakeholders do you invited for the discussion? You can mention the relevance of application of focus group for the study.
Results and Discussions
Within this section you should refer to the action of at least one external shock sinthetically in order to corespond to the Special issue "Smallholder Farming under External Shocks: New Perspectives and Solutions for Future Crises".
You mentioned that you conducted the study between 2016-2017. One new external shock, for example Covid-19 Pandemic, was occurred in the meantime. Is is possible that this factor to generate a more difficult situation in the study area ?
At the end of this section, the discussions should includ: a sinthetically presentation of the significance of the results highligthing their importance and the originality of the study.
Conclusions:
You intitled this section "Discussions and conclusion". You named the previous section "Results and Discussions". The fist part of this section contains ideas that can be added to the end of the previous section. It is better to separate the conclusions and discussions. Thus, the discussions should remain in the results section. No citations are used in the conclusions section.
You should mentioned the limitation of the study and future research at the end of conclusions.
Author Response
Reveiwer-2
I consider that the article addresses an interesting topic related to the analysis of role of various welfare and development schemes implemented by Indian government to reduce poverty among the population of the Kumkudupamula village.
Introduction:
The aim of the study is mentioned on the literature survey section. You should include it at the end of introduction.
Included
Methodology:
Description of study area: the maps of study village should be inserted in this subsection. I think that the figure 1B is not relevant in the sense that it does not provide aditional information about the location of the village. Moreover, the figure 1b is unclear. If you have the ability to elaborate a map that reflects land use, it would be best to include it-this is a proposal, not a mandatory suggestion.
Map removed
Can you argue why you choose this case study?
Yes, included in introduction and methodology
You should present several details regarding the manner of application of the semi-structured questionnaires: Who submitted the questionnaires? The period of application; did you use a model of the questionaire from other studies or do you make a specific structure in relation with the characteristics of the case study?
Yes, explained
You refer also to the method focus group but there are no details about the implementation of this method. Which stakeholders do you invited for the discussion? You can mention the relevance of application of focus group for the study.
Yes, included
Results and Discussions
Within this section you should refer to the action of at least one external shock sinthetically in order to corespond to the Special issue "Smallholder Farming under External Shocks: New Perspectives and Solutions for Future Crises".
You mentioned that you conducted the study between 2016-2017. One new external shock, for example Covid-19 Pandemic, was occurred in the meantime. Is is possible that this factor to generate a more difficult situation in the study area ?
At the end of this section, the discussions should includ: a sinthetically presentation of the significance of the results highligthing their importance and the originality of the study.
COVID related issues included
Conclusions:
You intitled this section "Discussions and conclusion". You named the previous section "Results and Discussions". The fist part of this section contains ideas that can be added to the end of the previous section. It is better to separate the conclusions and discussions. Thus, the discussions should remain in the results section. No citations are used in the conclusions section.
Separated
You should mentioned the limitation of the study and future research at the end of conclusions.
Yes, mentioned

Reviewer 3 Report
The papers shows significant results about the support/subsidies given in the rural areas of India, proving that these ones are not reached by the poorest of the society. Some minor changes must be done:
1. Introduce the graphic scale and north symbol in maps.
2. Remove the figure 1B. It lacks of north symbol, scale, and also, its quality is low.
3. Mention all the figures along the text, and always before of these.
4. Some index showing the inequalities could be added: Lorenz Curve or Gini Index.
5. For Table 3, must be introduced a paragraph explaining the main features of this table.
6. "Framers", in table 3, I guess will be "farmers"
7. Extend the Discussion and conclusion paragraph, with possible solutions to solve these inequalities and different access to subsidies, and possible following research.
Author Response
Reviewer 3
The papers shows significant results about the support/subsidies given in the rural areas of India, proving that these ones are not reached by the poorest of the society. Some minor changes must be done:
- Introduce the graphic scale and north symbol in maps.
Yes, included
- Remove the figure 1B. It lacks of north symbol, scale, and also, its quality is low.
Removed
- Mention all the figures along the text, and always before of these.
Done
- Some index showing the inequalities could be added: Lorenz Curve or Gini Index.
Yes included Lorenz curve
- For Table 3, must be introduced a paragraph explaining the main features of this table.
We have already having similar graphs, hence not done
- "Framers", in table 3, I guess will be "farmers"
Corrected
- Extend the Discussion and conclusion paragraph, with possible solutions to solve these inequalities and different access to subsidies, and possible following research.
Done

Reviewer 4 Report
Dear authors,
The present study is addressing the very important subject of theoutreach of welfare and development schemes in India, a country which faces extreme levels of poverty in many areas. The results of the study could provide improtant lessons not only for the country, but also for other areas that are facing similar issues.
The Results part of the manuscript is well written and comprehensive. Howwever I do have a few concerns regarding the research design that I think must be addressed before its publication.
1. The researched area is only one village with a population of 903. Can the results be generalized fot the entire country? If not, I think that is improtant to adress this limitation.
2. Another limitation that must be addressed is the time frame. The study was conducted during the 2016-2017 period. Are the results still up to date? Since then important events unfolded, such as the Sars-Cov-2 virus pandemic. Have these events affected the policies discussed in the manuscript?
All the best
Author Response
Reveiwer-4
The present study is addressing the very important subject of the outreach of welfare and development schemes in India, a country which faces extreme levels of poverty in many areas. The results of the study could provide important lessons not only for the country, but also for other areas that are facing similar issues.
The Results part of the manuscript is well written and comprehensive. However I do have a few concerns regarding the research design that I think must be addressed before its publication.
- The researched area is only one village with a population of 903. Can the results be generalized fot the entire country? If not, I think that is important to adress this limitation.
We have selected an average village which more or less represent India. Off course, it is one of the limitation of the study, but this case study will give holistic understanding of the village economy and welfare schemes.
- Another limitation that must be addressed is the time frame. The study was conducted during the 2016-2017 period. Are the results still up to date? Since then important events unfolded, such as the Sars-Cov-2 virus pandemic. Have these events affected the policies discussed in the manuscript?
We have included data up to 2022 in the revised version and highlighted COVID situation and changes in budget allocation and reach by taking example of Direct Benefit Transfers.

Round 2
Reviewer 1 Report
Thank the author for his efforts in revising the article. However, to be frank, I still do not understand the logic of the author's story. The author seems to be trying to evaluate the effect of the policy implementation, but it is impossible to accurately evaluate the effect of the policy only through simple descriptive statistical analysis and group comparison. At the same time, there are too many contents to present in the whole study, which makes it difficult for readers to follow the author's logic.
Reviewer 4 Report
Dear authors,
I think that the changes are in line with my obervation.
All the best!